# Revisiting Reinforcement Learning for LLM Reasoning from A Cross-Domain Perspective

Zhoujun Cheng[1,*], Shibo Hao[1,*], Tianyang Liu[1,*]
Fan Zhou[2], Yutao Xie[1], Feng Yao[1], Yuexin Bian[1], Yonghao Zhuang[3], Nilabjo Dey[4]
Yuheng Zha[1], Yi Gu[1], Kun Zhou[1], Yuqi Wang[2], Yuan Li[3], Richard Fan[2], Jianshu She[2]
Chengqian Gao[2], Abulhair Saparov[4], Haonan Li[2], Taylor W. Killian[2], Mikhail Yurochkin[2]
Zhengzhong Liu[2], Eric P. Xing[2,3], Zhiting Hu[1]

[1]UC San Diego    [2]MBZUAI    [3]Carnegie Mellon University    [4]Purdue University
[*]*Equal Contribution*

## Abstract

Reinforcement learning (RL) has shown promise in enhancing large language model (LLM) reasoning, yet progress towards broader capabilities is limited by the availability of high-quality, multi-domain datasets. This work introduces GURU, a 92K RL-for-reasoning dataset designed to address this gap, covering six reasoning domains: Math, Code, Science, Logic, Simulation, and Tabular, each with corresponding verifiers. We build GURU via a careful data-curation pipeline, including sourcing, deduplication, reward design, and domain-specific and difficulty-based filtering. With GURU, we present a systematic investigation of cross-domain RL generalization, and reveal several key aspects affecting cross-domain transferability. We further train two models GURU-7B and GURU-32B purely with RL on our curated data and observe largely improved performance over leading open RL reasoning model baselines, with gains of 7.3% and 7.8% respectively on an extensive 17-task, six-domain evaluation suite. We are releasing our dataset, code, and evaluation suite to the community, aiming to support further research and development of more general RL-enhanced reasoning models.

## 1 Introduction

Recent frontier reasoning models trained with reinforcement learning (RL), such as OpenAI-o1 [OpenAI, 2024] and DeepSeek-R1 [Guo et al., 2025], demonstrate impressive performance across diverse reasoning tasks. While many open-source efforts have attempted to unveil successful RL strategies, many of the analysis [Yue et al., 2025, Zeng et al., 2025] and training recipes [Hu et al., 2025, Luo et al., 2025b, He et al., 2025, Yu et al., 2025] are constrained to Math and Code domains. However, whether or not these models can be reliably extended to other domains has not been adequately established. This leaves two key questions for the community: *across a broader spectrum of reasoning challenges*, **(1)** to what extent do RL-enhanced reasoning abilities transfer between diverse domains, and **(2)** how can we build models that maintain high performance across these domains?

A major obstacle to tackling these questions is the lack of suitable data. Unlike supervised fine-tuning (SFT), where a model receives token-level supervision, improving reasoning with RL heavily relies on the quality of reward signal. As a result, public RL reasoning datasets are skewed toward competition problems which are accompanied by reference solutions or unit tests that offer a high-quality verification signal. Recent attempts to mine reward signals at scale from the web [Yuan et al., 2025, Ma et al., 2025] have not yet achieved the consistent quality needed to translate into clear improvements in general reasoning using RL.

39th Conference on Neural Information Processing Systems (NeurIPS 2025) Track on Datasets and Benchmarks.

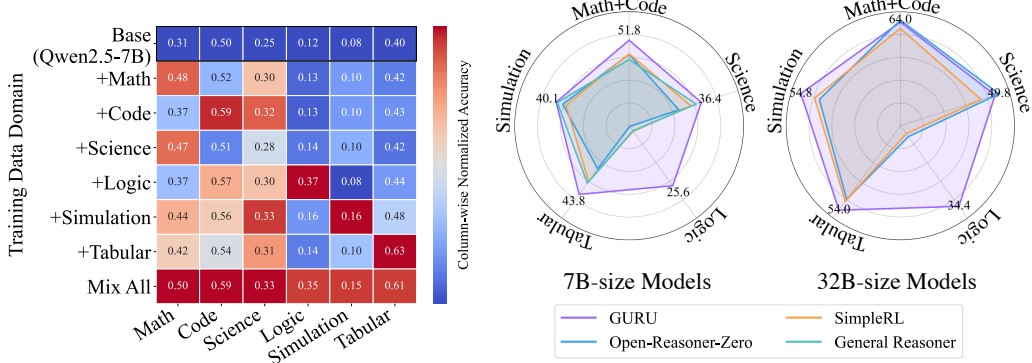

(a) Cross-domain transfer performance.

(b) Model performance across reasoning domains.

Figure 1: **Left:** Heatmap illustrating the transferability of domain-specific training. The best validation accuracy (%) achieved per domain during RL training is presented in each cell. **Right:** Comparison of 7B-scale/32B-scale models across the reasoning categories. Our model (GURU) consistently outperforms strong baselines in most domains, demonstrating superior generalization and balanced reasoning capability.

We introduce GURU, an open, curated RL corpus that spans six reasoning domains—Math, Code, Science, Logic, Simulation, and Tabular. Each domain undergoes a five-stage construction pipeline: (1) *Data sourcing*: collect high-quality public datasets or synthesize new ones. (2) *Deduplication*: eliminate near-duplicate problems to ensure diversity. (3) *Reward design*: build rule-based, execution-based, or model-based verifiers that produce reliable rewards. (4) *Heuristic filtering*: remove noisy or trivial examples using domain insights and pilot studies. (5) *Difficulty filtering*: remove noisy or unstable samples and retain challenging ones to enhance data quality and focus RL training on informative examples. The resulting corpus contains 92k examples, each paired with a reliable reward function ready for RL research.

Building upon the multi-domain RL data from GURU, we conduct a series of controlled experiments using the Qwen2.5-7B [Yang et al., 2024] model to systematically analyze the domain generalization of RL training. Our analysis yields several interesting findings: (1) Math, Code, and Science exhibits substantial performance gains from cross-domain RL training, markedly exceeding transfer seen in other domains. (2) Training on a simple mixed-domain corpus proves highly effective, consistently achieving performance comparable to or surpassing models trained solely on single-domain data. See Figure 1 (left) for illustrations. (3) The training dynamic can be domain-dependent, e.g., only Math and Code show increased response length during RL, and the joint training can influence the dynamic on specific domains. (4) While training on harder examples can boost in-domain performance, it carries the risk of degrading performance on cross-domain tasks.

Finally, we deliver GURU-7B and GURU-32B, two general reasoning models trained with RL on our dataset without SFT. On our unified evaluation suite with 17 reasoning tasks across six domains, our models establish a new state-of-the-art within the category of open models trained with publicly available data using RL. Specifically, GURU-7B surpasses Open-Reasoner-Zero-7B [Hu et al., 2025], by an average of 7.3%, and GURU-32B outperforms Open-Reasoner-Zero-32B by 7.8%. This significant performance leap on a general reasoning suite highlights a crucial gap in multi-domain capabilities compared to prior open efforts that have often seen stronger performance concentrated in single domains like Math and Code. Our results demonstrate the efficacy of the GURU data in advancing the frontier of general reasoning performance within the open ecosystem. To catalyze further research and accelerate progress towards truly general reasoning, we would later release our series of GURU reasoning models, the GURU dataset, the evaluation suite, and the corresponding code. This is done to encourage the community to move beyond siloed single-domain pursuits and collaboratively explore the challenges of more general, multi-domain reasoning.

## 2 Data Construction

Datasets used for Reinforcement Learning from Verifiable Rewards (RLVR) [Lambert et al., 2024] predominantly focus on narrow domains such as Math [Luo et al., 2025b, Hu et al., 2025, He et al., 2025, Zeng et al., 2025] and Code [Liu and Zhang, 2025, Luo et al., 2025a]. However, applying RL on single-domain datasets often results in overfitting to the specific structures and heuristics of that domain. As shown in Figure 1, models trained to excel on advanced math benchmarks exhibit sharp performance drops when evaluated other domains. Even datasets that claim to cover broad reasoning capabilities tend to remain confined within STEM boundaries and experience similar generalization failures. Moreover, current RLVR datasets often suffer from high redundancy, noisy queries, and poor difficulty calibration. These issues call for principled data curation, including deduplication, domain-specific filtering, and difficulty screening. To address these challenges, we construct a comprehensive pipeline for building GURU, a multi-domain, reward-verifiable dataset designed to improve general reasoning through RL.

### 2.1 Data Pipeline

Figure 2 presents an overview of our data curation pipeline, which is designed to ensure both domain diversity and reward verifiability, and consists of the following five stages:

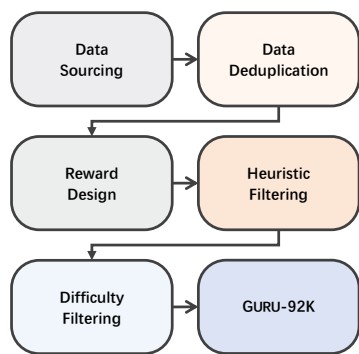

Figure 2: Overview of the data curation pipeline of GURU dataset.

**Data Sourcing** We curate data across six reasoning-intensive domains, ensuring both diversity and verifiability: **(1) Math:** we adopt recent math reasoning collection *OR1* [He et al., 2025], *DAPO* [Yu et al., 2025], and *DeepScaler* [Luo et al., 2025b], which compile numerous Math datasets, including previous competition problems like AIME or AMC. **(2) Code:** we include programming problems sourced from online coding platforms, programming competitions, and synthetic code generation tasks. Our collection includes real-world problems from *LeetCode* [Xia et al., 2025], curated and verified problems from *TACO-Verified* [Li, 2024], synthetic tasks from *PrimeIntellect* [Mattern et al., 2025], and historical problems from *LiveCodeBench* [Jain et al., 2024]. Notably, we reuse filtered subsets of *PrimeIntellect* and *LiveCodeBench* processed by *DeepCoder* [Luo et al., 2025a]. **(3) Science:** we include *WebInstruct-Verified* [Ma et al., 2025], a dataset crawled from web and processed with LLMs as the data source for science. **(4) Logic:** we include symbolic reasoning tasks sourced from both existing datasets and our own synthetic tasks. For existing datasets, we include *ARC-AGI*(1 and 2) [Chollet et al., 2024], focusing on inducing abstract rules over symbolic grids; and *BARC* [Li et al., 2024b], which extends these tasks for inductive and transductive generalization. Additionally, we synthesize three tasks for RL training: *Zebra Puzzle* [Lin et al., 2025], a classic Logic grid puzzle requiring models to chain positional and equality constraints; *Ordering Puzzles*, requiring models to recover the unique linear order of up to 50 objects from diverse relational constraints; and *Graph Search*, where models must construct a predicate graph from natural language facts and reason over it to find entity paths. **(5) Simulation:** we include simulation-style reasoning tasks from *Code I/O (PyEdu)* [Li et al., 2025], where models must predict outputs or infer inputs based on code without actual execution. **(6) Tabular:** we repurpose existing table-based question answering datasets involving single- and multi-table reasoning. Specifically, we adopt *HiTab* [Cheng et al., 2021] and *MultiHierTT* [Zhao et al., 2022]. Table 1 summarizes the data sources and sample scales for each domain. Further details are provided in Appendix.

**Data Deduplication** Significant content overlap exists within Math and Code domains, as many datasets originate from shared upstream sources. For instance, many Math datasets reuse problems from past AIME exams or curated collections like NuminaMath [Li et al., 2024a]. Similarly, Code datasets are often derived from online coding platforms (e.g., LeetCode) and curated benchmarks (e.g., APPS, CodeContests), leading to repeated problems across datasets. In our preliminary experiments, we find similarity-based metrics such as embedding distance, Jaccard similarity, or n-gram overlap, are prone to false positives. E.g., n-gram checking may incorrectly flag distinct samples that share a common prefix or template but diverge in the actual problem content. To address this, we perform deduplication using a relatively conservative matching strategy. If one question is a strict substring of

Table 1: Dataset statistics across all 6 domains, including raw sizes and filtered subsets.

| Domain | Dataset | Raw Size | w/ Dedup. & Domain Filt. | w/ Difficulty Filt. |
|---|---|---|---|---|
| **Math** | OR1 [He et al., 2025]
DAPO [Yu et al., 2025]
DeepScaler [Luo et al., 2025b] | 105k
17k
40.3k | 118.2k
(-27.2%) | 54.4k
(-54.0%) |
| **Code** | LeetCode [Xia et al., 2025]
TACO-Verified [Li, 2024]
PrimeIntellect [Mattern et al., 2025]
LiveCodeBench (history) [Jain et al., 2024] | 2k
12.9k
16.3k
0.9k | 23.7k
(-26.2%) | 18.1k
(-23.6%) |
| **Science** | WebInstruct-Verifed [Ma et al., 2025] | 232k | 31.7k
(-86.3%) | 3.6k
(-88.6%) |
| **Logic** | Zebra Puzzle
Ordering Puzzle
Graph Puzzle
ARC-AGI [Chollet et al., 2024]
ARC-AGI-2 [Chollet et al., 2025]
BARC [Li et al., 2024b] | 5.7k
2.9k
2.8k
0.4k
1k
3.4k | 13.6k
(-12.8%) | 6.3k
(-53.7%) |
| **Simulation** | Code I/O (PyEdu) [Li et al., 2025] | 227k | 12.1k
(-94.7%) | 3.7k
(-69.4%) |
| **Tabular** | HiTab [Cheng et al., 2021]
MultiHierTT [Zhao et al., 2022] | 7.5k
7.8k | 10.3k
(-36.7%) | 6.1k
(-32.7%) |
| **Total** | – | **684.9k** | **209.6k** | **91.9k** |

another, the shorter sample is removed. This process removes 27.2% of math samples and 7.5% of code samples. To further support data quality assessment, we maintain a metadata-level mapping of dataset provenance and cross-source dependencies, detailed in Appendix.

**Reward Design**   A key challenge in RL for reasoning tasks is to design reward functions that are fully automatic, low-noise, and domain-appropriate. Across all domains in our dataset, we adopt binary rewards—a sample receives a reward of 1 if the output is judged as correct under domain-specific verification rules, otherwise 0. We categorize reward design into three types:

1. **Rule-Based Matching** is the most common strategy, used in domains such as Math, Logic, Simulation, and Tabular reasoning. Despite surface differences, these tasks share a common pattern: the model is prompted to output its final answer in a structured form—typically enclosed in a \box{}, a special tag (e.g., <answer>), or a JSON code block. The verifier extracts this region, normalizes the output, and applies a strict match. For math, we incorporate a symbolic program [Cui et al., 2025] to account for variations in expression.

2. **Execution-Based Verification** is used in Code domains, where correctness is defined by program behavior. The model generates a function or script, which is executed in a sandboxed environment against test cases. The reward is 1 only if all tests pass. For tasks that rely on stdin/stdout formats, we include fuzzy comparison routines to accommodate formatting and numerical variations.

3. **Model-Based Verification** is applied for Science, where answers are often open-ended and cannot be reliably checked with hard rules. Here, we use the 1.5B verifier model [Ma et al., 2025] to evaluate whether the model's output entails the reference answer. This allows us to scale RL to tasks that demand more semantic flexibility.

**Heuristic Filtering**   Reliable reinforcement learning critically depends on robust and accurate reward signals, which can be compromised by noisy, ambiguous, or poorly constructed training samples. To address this, we apply a systematic heuristic filtering stage across all domains to improve reward diversity, verifiability, and stability. We begin by removing samples with excessively long prompts, as input truncation due to token limits can strip essential information and hinder model performance. In the **Code** domain, we discard any sample whose reference solution fails its own unit tests. To ensure stable reward computation in high-concurrency, multi-process environments, we also filter out overly large inputs (e.g., stdin > 1MB) and randomly sample at most 8 test cases per example. For **Simulation** tasks, where multiple input-output variants may exist per program, we retain only a single pair per problem to maximize coverage and minimize redundancy. In **Logic** domain, we control task complexity: for ordering puzzles, we retain those with more than 20 objects; for graph-search tasks, we filter by retaining problems with node counts larger than 10 and higher

lookahead requirements; and for zebra puzzles, we preserve only samples with larger grids ($\geq 10$ objects, 5 attributes), to ensure robust symbolic and multi-step reasoning demands. In the **Science** domain, we restrict the data to university- and PhD-level physics, chemistry, and biology questions, excluding boolean and multiple-choice formats to avoid shortcut solutions. We also remove samples with excessively high numeric precision, which often result in brittle or unreliable reward evaluations.

**Difficulty Filtering**   To further enhance data quality and ensure that RL focuses on challenging and reliable samples, we introduce a *difficulty-aware filtering stage*. This step is designed to (i) prioritize samples that present sufficient difficulty to the model, and (ii) aggressively filter out samples that exhibit signs of annotation noise or unstable reward signals.

We implement this by measuring the empirical pass rates of both a weak model ($M_{\text{weak}}$, `Qwen2.5-7B-Instruct`) and a strong model ($M_{\text{strong}}$, `Qwen3-30B-A8B`) over $N = 16$ runs per sample, denoted as $P_{\text{weak}}$ and $P_{\text{strong}}$, respectively. Based on these statistics, we filter out samples that satisfy any of the following conditions:

1. **Overly easy samples**: $P_{\text{weak}} \geq \frac{15}{16}$. These examples are solved consistently by the weak model and thus offer little room for further reasoning improvements via RL.
2. **Potentially noisy samples**: $P_{\text{strong}} = 0$. We aggressively remove such samples, as consistent failure by the strong model suggests it may be ambiguous, malformed, or otherwise unreliable.
3. **Anomalous samples**: $P_{\text{weak}} > P_{\text{strong}}$. Empirical observations indicate that these samples often exhibit reward inconsistencies or memorization artifacts, where the weak model converges to an incorrect label but consistent output pattern that the strong model avoids.

To further refine the dataset in specific domains, we analyze the *difficulty gap*, defined as $P_{\text{strong}} - P_{\text{weak}}$. This metric serves as a proxy for the *learnability* of a sample, where higher values indicate greater potential for RL to improve model reasoning capabilities. In the Math domain, given its large scale and generally high baseline performance, we adopt a more aggressive filtering strategy. We remove samples where the difficulty gap is marginal ($P_{\text{strong}} - P_{\text{weak}} \leq \frac{6}{16}$) and the strong model already demonstrates high competence ($P_{\text{strong}} \geq 0.75$). This ensures that RL focuses on sufficiently challenging math problems where meaningful improvements remain achievable. In the Science domain, where data quality and model-based verifier are inherently noisier, we apply a stricter criterion by discarding samples with $P_{\text{strong}} - P_{\text{weak}} < 0.5$, prioritizing samples with clearer reasoning gaps and minimizing the inclusion of ambiguous or low-signal examples.

# 3   Analysis of Cross-Domain Reasoning Transfer

To gain a deeper understanding of how reasoning capabilities generalize across different domains with RL, we conducted a controlled experimental analysis with our GURU dataset. Specifically, we investigated the impact of RL on single reasoning domains versus a mixed-domain corpus on their subsequent performance across a suite of domain-specific benchmarks. These experiments aim to inform the design of additional large-scale RL reasoning experiments while also providing a broadened perspective on the utility of data diversity when developing reasoning models.

To this end, we constructed an experimental dataset by randomly sampling a subset of 3K training samples from each of the six domains covered by our GURU dataset. These subsets form a mixed training dataset totaling 18K samples called GURU-18K. We train reasoning models directly from the Qwen2.5-7B-Base model using RL on each of these single-domain subsets and the combined mixed dataset with a training batch size 512 and gradient update batch size 64 per step. Figure 3 illustrates the cross-domain generalization performance trained on every single domain and mixed (rows) and evaluated on various target domain tasks (columns). Our analysis yields several key findings.

## 3.1   Differential Transferability

**Domain identity matters.** Math, Code, and Science benchmark performance is consistently and significantly improved from other domains (Figure 3). This initially seems counter-intuitive, given that Math, Code, and Science are often considered highly complex reasoning tasks. We hypothesize this phenomenon stems from the base model's extensive exposure to math and code tokens during pretraining [Liu et al., 2024, Yang et al., 2024]. The model thus contains deep knowledge in these

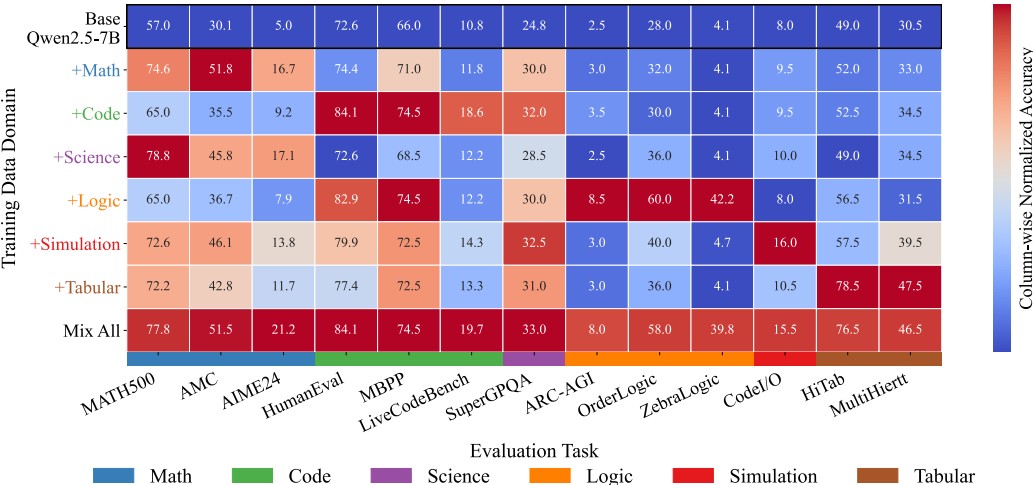

Figure 3: **RL Model Cross-Domain Transfer Performance.** The heatmap illustrates the normalized performance gains of models trained via RL on different domain configurations (rows: single domains or mixed corpus) when evaluated on the test sets of various domains (columns). Warmer colors indicate higher performance gains, computed by applying min-max normalization to the validation accuracies within each column. The best validation accuracy (%) achieved per domain during RL training is annotated in each cell. This highlights differential transferability: Math, Code, and Science benefit significantly from cross-domain transfer, while Logic, Simulation, and Tabular tasks see limited gains, with improvements primarily driven by within-domain training.

domains which is effectively elicited and refined through RL, even when the training data originates from different domains. Conversely, other domains exhibit limited or no substantial performance gains when trained on cross-domain data. This asymmetry highlights the critical importance of curating diverse cross-domain training data, particularly for domains less represented during pretraining, to achieve broad reasoning competence.

**Task difficulty matters.** As shown in Figure 3, easier tasks within Math (e.g., MATH500 [Hendrycks et al., 2021], AMC) and Code (e.g., HumanEval [Chen et al., 2021a], MBPP [Austin et al., 2021]) readily exhibit positive transfer from other domains. In contrast, performance on more challenging benchmarks in those same domains, namely AIME24 and LiveCodeBench [Jain et al., 2024], show considerably less improvement from cross-domain training. For the most challenging tasks, those with the lowest baseline absolute scores (e.g., ARC-AGI [Chollet et al., 2024], CodeI/O [Li et al., 2025]), we observe marginal to negligible gains from training on non-native domain data. This variance in reasoning transferability suggests that cross-domain RL alone is not sufficient to effectively develop models with general, complex reasoning capability. Thus, advanced competence in multiple domains is dependent on training with domain-specific examples or with substantially more diverse data, perhaps drawing from multiple difficult domains.

**Mixed-domain training matches or exceeds single-domain performance.** RL training on a uniformly mixed dataset of reasoning tasks proves remarkably effective as shown in the last row of Figure 3. Across individual downstream benchmarks, models trained on this combined data achieve performance levels consistently comparable to, and sometimes surpassing, those attained by models trained exclusively on data from the target domain. This demonstrates that even a simple uniformly mixed dataset across multiple domains can significantly enhance general reasoning capabilities with minimal apparent interference between the six domains involved. However, future research should investigate whether interference remains negligible as the number and diversity of included domains is increased. Scaling in this manner might necessitate more refined domain balancing strategies.

## 3.2 Reward and Response-Length Dynamics

Figure 4 shows the reward and average response length across six reasoning domains during RL fine-tuning. The upper row indicates RL training with 3k single-domain examples, while the lower

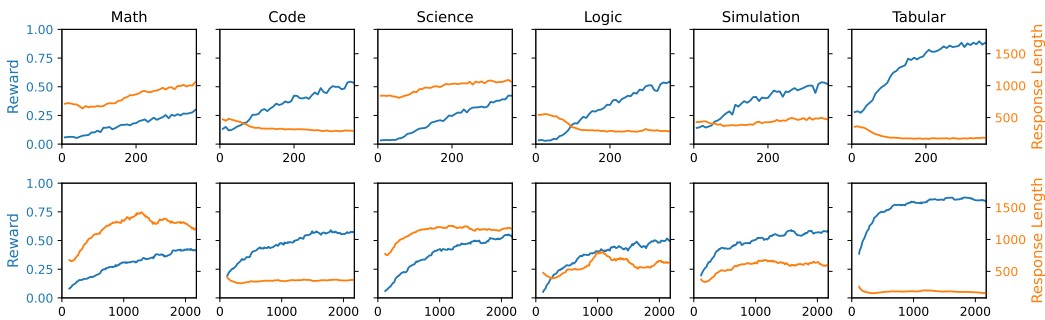

Figure 4: The reward and response length of each domain during RL training with: (**top row**) single domain data (3k examples each) from GURU-18k; and (**bottom row**): using the full GURU-18k mixture dataset. The x-axis is the number of gradient update steps.

Table 2: Performance (best validation accuracy within 200 RL steps) comparison trained on difficulty-filtered (harder) v.s. unfiltered (easier) math data. Training on harder data improves in-domain performance but leads to degradation on easier cross-domain tasks (e.g., HumanEval and HiTab).

| | Math (in-domain) | | | Code & Tabular (cross-domain) | | | |
| --- | --- | --- | --- | --- | --- | --- | --- |
| | **MATH500** | **AMC** | **AIME24** | **HumanEval** | **LiveCodeBench** | **HiTab** | **Multihiertt** |
| **Unfiltered math** | 75.8 | 52.1 | 15.8 | 82.3 | 11.1 | 56.5 | 32.0 |
| **Difficulty-filtered math** | 78.6 | 58.4 | 21.7 | 73.1 | 10.7 | 53.5 | 35.5 |
| $\Delta$ (+/-) | +2.8 | +6.3 | +5.9 | -9.2 | -0.4 | -3.0 | +3.5 |

row indicates joint training with 18k examples. The x-axes are aligned so that each horizontal position represents an equal number of target domain samples.

In single-domain training, contrary to the common belief that RL drives models to produce longer responses, we observe strong domain effects: on Code, Logic, and Tabular tasks, the policy actually contracts its outputs, while Science and Math become more verbose, and simulation remains largely unchanged. When we switch from single-domain training to joint training on the full GURU -18k mixture, rewards climb steeply within the first few hundred gradient steps across all domains. This demonstrates positive cross-domain transfer. Joint training may also reshape length dynamics: for Logic, the single-domain run monotonically shortens answers, but the multi-domain run first lengthens and then shortens them, suggesting that shared representations learned from other domains modulate brevity preferences. In contrast, domains such as Tabular preserve their "short-answer" tendency even under joint training, indicating that some length priors remain robust to cross-domain influence.

### 3.3 Effects of Training Data Difficulty

We further conducted an ablation to investigate the impact of training data difficulty on both in-domain performance and cross-domain transferability. Specifically, we trained the base Qwen2.5-7B-Base model using RL for 200 steps on either the complete, unfiltered Math domain data (representing a mix of difficulties) or a difficulty-filtered subset containing primarily harder Math problems from GURU. Other training configurations followed Section 3.1. We then evaluated performance on the Math tasks (in-domain) and selected Code and Tabular analysis tasks (cross-domain).

As shown in Table 2, training on difficulty-filtered math data consistently improves in-domain performance compared to unfiltered data, particularly on harder Math tasks like AMC (+6.3) and AIME24 (+5.9). However, the effects on cross-domain tasks are difficulty-dependent. Easier tasks such as HumanEval (-9.2) and HiTab (-3.0) suffer notable degradation. This decline was corroborated by observed accuracy collapse (large drops after approximately 90 steps and 150 steps respectively) during training on the filtered data. Conversely, performance on harder cross-domain tasks (LiveCodeBench, Multihiertt) shows minimal negative impact or positive change.

In summary, increasing training data difficulty within a domain can consistently enhance in-domain performance. It is reasonable to perform aggressive difficulty filtering when the sole objective is maximizing performance within that specific domain on challenging tasks, e.g., AIME and

Table 3: Full 17 benchmark performance on 7B and 32B Models. GURU outperforms Baseline Models with notable improvements. ◇: ORZ represents Open-Reasoner-Zero [Hu et al., 2025].

| Models →
Benchmarks ↓ | | GURU
7B | General
Reasoner
7B | SimpleRL
Qwen-2.5
7B | ORZ
7B◇ | GURU
32B | ORZ
32B◇ | SimpleRL
Qwen-2.5
32B |
|---|---|---|---|---|---|---|---|---|
| **Math** | AIME24 | 17.50 | 17.08 | 15.60 | 16.25 | 32.29 | 47.50 | 27.20 |
| | MATH500 | 77.25 | 70.40 | 87.00 | 80.80 | 84.00 | 89.80 | 89.60 |
| **Code** | LiveCodeBench | 16.49 | 8.24 | 7.21 | 5.73 | 28.32 | 22.93 | 19.35 |
| | HumanEval | 78.45 | 64.02 | 58.61 | 63.41 | 91.16 | 84.14 | 82.24 |
| | MBPP | 69.35 | 40.60 | 49.25 | 50.05 | 78.45 | 75.80 | 76.75 |
| **Science** | GPQA-diamond | 40.90 | 36.87 | 34.85 | 20.20 | 52.02 | 53.53 | 43.94 |
| | SuperGPQA | 31.80 | 30.64 | 27.29 | 29.75 | 43.60 | 46.05 | 37.73 |
| **Logic** | ARC-AGI | 7.00 | 1.34 | 0.90 | 0.00 | 18.50 | 13.00 | 4.50 |
| | Zebra Puzzle | 29.43 | 0.00 | 1.00 | 1.00 | 31.50 | 1.00 | 0.50 |
| **Simulation** | CodeI/O | 13.00 | 8.50 | 7.00 | 1.00 | 14.00 | 2.50 | 12.00 |
| | CruxEval-I | 61.72 | 63.63 | 56.25 | 71.13 | 79.38 | 71.13 | 72.63 |
| | CruxEval-O | 71.28 | 56.50 | 58.31 | 64.88 | 88.38 | 82.38 | 67.75 |
| **Tabular** | FinQA | 34.70 | 34.33 | 35.10 | 15.34 | 45.32 | 45.20 | 45.41 |
| | HiTab | 52.00 | 42.50 | 36.00 | 37.50 | 60.00 | 44.00 | 47.00 |
| | MultiHiertt | 44.72 | 33.04 | 35.42 | 29.76 | 56.55 | 52.08 | 52.68 |
| **Others** | IFEval | 35.81 | 39.56 | 36.69 | 32.72 | 58.04 | 38.26 | 55.27 |
| | LiveBench | 18.57 | 29.12 | 15.20 | 12.64 | 34.03 | 28.78 | 28.33 |
| **Average Score** | | **41.17** | 31.30 | 33.04 | 33.90 | **52.68** | 46.95 | 44.88 |

LiveCodeBench. While for cross-domain transfer, it introduces a risk of negative transfer to easier tasks in other domains. These results suggest that for beneficial cross-domain transfer, a more balanced distribution of training data difficulties, or the explicit inclusion of cross-domain data, may be more effective than solely increasing the difficulty of the source domain data.

# 4 Main Experiment

Having motivated the need for multi-domain data in cultivating general reasoning skills (Section 3), we pivot to demonstrating its practical impact through large-scale RL training. This section reports on training 7B and 32B models on the full GURU dataset. We empirically show that mixed multi-domain training is indeed effective at scale. As focus on this fundamental demonstration, we leave fine-grained studies on data scaling effects and online data mixing for future work. Section 4.1 outlines our experimental approach, followed by a comprehensive presentation of results in Section 4.2.

## 4.1 Experimental Setup

**Training Configurations**   We use `verl` [Sheng et al., 2024] as the RL training framework and GRPO [Shao et al., 2024] as the RL training algorithm. For general training hyper-parameters, we utilize the AdamW optimizer [Loshchilov and Hutter, 2017], incorporating a linear warm-up of 10 RL steps. The prompt batch size is 512 for one RL step, and we sample 16 responses for each prompt with a temperature of 1.0. The mini-batch size is set to 64, i.e., 8 gradient updates for each RL step. The maximum number of tokens is 4k for input prompt and 8k for generation. All experiments are conducted on 24 GPU nodes, each equipped with 8 Hopper GPUs, for both RL training and evaluation. The 7B model was trained on 4 nodes for 2 epochs from Qwen2.5-7B-Base and the 32B model was trained on 16 nodes for 2 epochs from Qwen2.5-32B-Base, each lasting for 2.5 days.

**Baselines**   For baseline comparisons, we select the most performant RL-trained reasoning models with open data in math and general reasoning at 7B and 32B scale: (1) Open-Reasoner-Zero (ORZ) [Hu et al., 2025], (2) SimpleRL-Zoo [Zeng et al., 2025], and (3) General Reasoner [Ma et al., 2025] We exclude distilled models from baselines for a fair com-

parison with our models RL-trained from base models without SFT. Also, it can directly verify our GURU dataset's effectiveness without dependency on the SFT data.

**GURU Evaluation Suite**    Our evaluation covers 17 benchmarks across six key domains to assess model capabilities comprehensively. The principle behind our evaluation is to establish rigorous standards using challenging tasks that measure both the breadth and depth of reasoning. (1) **Math**: We evaluate on two mathematical competition benchmarks: AIME24 [MAA, 2024] and MATH500 [Hendrycks et al., 2021]. (2) **Code Generation**: We measure standard program synthesis using HumanEval [Chen et al., 2021a] and MBPP [Austin et al., 2021], while LiveCodeBench [Jain et al., 2024] provides dynamic evaluation of competitive programming skills. (3) **Science**: We use GPQA and the recently released SuperGPQA, which spans over 100 scientific disciplines and presents significantly increased difficulty. (4) **Logical Reasoning**: The suite includes ARC-AGI1 [Chollet et al., 2024] for abstract reasoning tasks, and our self-synthesized benchmark based on the classic Zebra Puzzle format [Lin et al., 2025] to evaluate logical deduction capabilities. (5) **Simulation**: We sample 200 examples from CodeI/O [Li et al., 2025] to assess interactive simulation and reasoning, and incorporate CRUXEval [Gu et al., 2024] for evaluating code-based reasoning capabilities. (6) **Tabular Tasks**: We've also verified model's tabular reasoning ability on Hitab [Cheng et al., 2021] and MultiHiertt [Zhao et al., 2022]'s test split, as well as a financial tabular reasoning benchmark, FinQA [Chen et al., 2021b]. (7) **Other Tasks**: We also include IFEval [Zhou et al., 2023] and LiveBench [White et al., 2024] to test performance on novel, unseen task types, further probing generalization and robustness. For scoring, ARC-AGI uses `Pass@8`, while all other benchmarks report `Avg@k` with $k \in \{1, 8, 32\}$. We only sample 32 times for AIME; any benchmarks with more than 500 examples will be sampled only once. All experiments adopt R1's inference settings with temperature `T=0.6` and `Top-p=0.95`.

## 4.2 Results

As shown in Table 3, GURU-7B and GURU-32B consistently demonstrate more balanced and advanced performance across all six skill set evaluations. For overall performance, GURU-7B achieves an average score of **41.17%**, outperforming the second-best RL model, Open-Reasoner-Zero-7B, by 7.3%. Similarly, GURU-32B attains **52.68%**, surpassing Open-Reasoner-Zero-32B by over 7.8%. Notably, even without explicit training on certain tasks, applying RL with GURU also enhances the model's generalization ability—showing strong results on novel task types such as IFEval. This clearly demonstrates the exceptional quality and effectiveness of the GURU dataset in promoting a wide scope of reasoning ability, highlighting its value as a strong foundation for building general-purpose reasoning models.

## 5   Related Work

Reinforcement Learning from Verifiable Rewards (RLVR) has emerged as a powerful paradigm for enhancing the reasoning capabilities of Large Language Models [Guo et al., 2025, OpenAI, 2024]. Following initial successes, a significant body of open work has explored RLVR, primarily concentrating on specializing models for highly challenging single domains. Efforts such as Open-Reasoner-Zero [Hu et al., 2025], Skywork-OR1 [He et al., 2025], DeepScaler [Luo et al., 2025b], and SimpleRL [Zeng et al., 2025] have notably leveraged extensive mathematical data to achieve state-of-the-art performance on complex math benchmarks. Similarly, DeepCoder [Luo et al., 2025a] focused on RL for code generation tasks. While powerful within their specific areas, this domain-specific focus inherently limits the generalizability of the resulting models across the broader landscape of reasoning tasks. Co-current works like General-Reasoner [Ma et al., 2025] and Nemotron-CrossThinker [Akter et al., 2025] have begun to explore broader domains for RL training. However, their scope remains constrained to STEM problems, leaving many crucial scenarios, including comprehensive coding, logic, and tabular analysis unexplored by large-scale open RL efforts. Addressing this critical gap in open resources and models for general reasoning via RLVR, we introduce GURU, a novel multi-domain dataset spanning six reasoning domains: Math, Code, Science, Logic, Simulation, and Tabular. Utilizing GURU, we train GURU-7B/32B, general reasoning models optimized via RL exclusively on open data using our GURU dataset and see state-of-the-art results on general reasoning among open models trained with publicly available data using RL.

# 6  Conclusion

In this work, we provide a curated dataset GURU for RL on general domains, covering Math, Code, Science, Logic, Simulation, and Tabular analysis. We perform a controlled experiment on how RL reasoning transfers across domains based on GURU and find out that domains and task difficulty affect transferability and mixed-domain training matches or exceeds single-domain performance. Finally, we developed GURU-7B and GURU-32B, general reasoning models optimized via RL on our multi-domain GURU dataset using only open data, demonstrating state-of-the-art performance among open models on an extensive reasoning evaluation suite over 17 tasks across six domains. These models show a notable leap in general reasoning performance, highlighting the gap left by previous open efforts concentrated on single domains like math or code. We would make the GURU dataset, models, evaluation suite, and code available, aiming to support the community's efforts towards more general, multi-domain reasoning research.

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

## Societal Impacts

The release of the GURU, its unified evaluation suite, and the accompanying reinforcement-learning code can substantially lower the barrier to entry for research on general reasoning. By providing a high-quality, multi-domain resource under a permissive open-source license, the work enables universities, non-profits, and smaller companies to reproduce and build upon state-of-the-art methods without relying on proprietary data. A standardized benchmark spanning mathematics, code, logic, tables, simulation, and science also helps the community expose domain-specific blind spots in current models and encourages the development of more balanced, robust, and transparent reasoning systems. In education, the automatically verifiable problems included in GURU can serve as rich material for coursework and online learning platforms.

At the same time, powerful cross-domain reasoning models pose several risks. High-accuracy generation of mathematical proofs or executable code could facilitate academic dishonesty or the automated creation of malicious scripts; the English-centric, STEM-heavy data may amplify linguistic and disciplinary biases; and large-scale RL training increases computational and energy costs. To mitigate these concerns we (i) provide verifier source code and safety-filter examples to foster downstream auditing and human-in-the-loop deployment, and (ii) restrict the corpus to a curated 92 k high-information samples so as to avoid unnecessary compute.

## Assets licenses

Our multi-domain corpus draws on a variety of public and synthetic datasets with differing license terms. LiveCodeBench and HiTab are released under the MIT and Apache 2.0 licenses, respectively, while TACO and WebInstruct-verified use Apache 2.0 and CC BY 4.0. The ARC-AGI (Challenge & Easy) benchmark is CC BY-SA, and the Graph Logical Dataset is CC BY 4.0. GURU incorporate a mix of MIT, CC BY 4.0, and Apache 2.0–licensed content.

