# OpenReview forum: "Revisiting Reinforcement Learning for LLM Reasoning from A Cross-Domain Perspective"
_NeurIPS.cc/2025/Datasets_and_Benchmarks_Track — NeurIPS 2025 Datasets and Benchmarks Track poster_

### Official Review · Reviewer_5khS · 2025-07-02

**Rating:** 5
**Confidence:** 5

**Summary:**

GURU is a strong step toward general RL LLMs—by providing a multi-domain, verified, open dataset and showing that RL training on such data leads to much better general reasoning skills than current open baselines. The work closes a key gap and enables further research in general reasoning.

**Dataset Code Accessibility:**

Yes

**Ethical Considerations:**

No, there are no or only very minor ethics concerns

**Final Justification:**

Most concerns have been solved

**Limitations Weaknesses:**

1. Baselines seem to be **unfair**. The current comparisons between models may not be fair. ORZ baseline is only trained on math-domain datasets, while the proposed method is trained on data from multiple domains. Comparing a model trained broadly across domains to one limited to a single domain (like MATH) can lead to misleading conclusions. For a fair and meaningful evaluation, the baseline models should also be trained on similar multi-domain data, or the proposed model should be tested using only math-domain data when compared with ORZ.

2. The results in Table 2 (best validation accuracy within 200 RL steps) show that harder datasets can accelerate training, but not necessarily improve final performance. This suggests that simple queries contribute less to learning. However, it remains unclear whether the performance has already stabilized within 200 steps.  There could still be improvements happening after that point, which are not shown in the table. I think the stable results/performance is required.

3. Base Model Choice. Qwen is proved to work well for the RL method (magic), but it would be better to test at least one other open-source model besides Qwen. This would help demonstrate that good performance obtained from RL with your data is not dependent on Qwen’s specific behavior and can generalize across LLMs.

These weaknesses can likely be fixed by adding more experimental results. I appreciate the useful training data. Just wish the paper could be stronger and give me more confidence in using these data.

**Strengths Contributions:**

1. Large, open, high-quality RL dataset covering six reasoning domains, not just Math and Code.
2. Enables better general reasoning in open-source LLMs.
3. Well-organized, clear writing; uses helpful figures/tables; thorough experimental detail.

---

> ### Author Rebuttal · Authors · 2025-07-31
>
> We sincerely thank you for your incredibly thoughtful and encouraging review. We are especially grateful for your positive summary recognizing Guru as a **"strong step"** that **"closes a key gap"** for general RL reasoning. Your clear articulation of the paper's strengths is very motivating.
>
> We also agree entirely with your assessment that the paper could be made even stronger with additional experiments to boost confidence. To that end, we have worked diligently to conduct **three new sets of experiments** that directly address each of the weaknesses you identified. We are excited to share these new results with you.
>
> ---
>
> > W1: The selection of baselines for comparison seems unfair.
> >
>
> Thank you for your constructive feedback. We have run extensive new experiments with **8 additional baselines** and **a new SOTA result compared with OpenAI o4-mini** to provide a clearer and more comprehensive evaluation of Guru.  Our evaluation strategy and new results are as follows:
>
> 1. **Baseline strategy.** Unlike frontier open-weight reasoning models like Qwen3, and close ones like OpenAI-o4-mini, our Guru models do not undergo SFT before RL. Data used in Qwen and OpenAI series are publicly unavailable. SFT significantly boosts performance and is different from our focused RL phase in this paper.
>     - **Direct comparison:** For this reason, General Reasoner is our most important baseline, as it is also trained for general reasoning via RL on open data. As shown in the original paper and with new models in Table 1, our Guru-7B model significantly outperforms General-Reasoner 7B, demonstrating the superior quality of our dataset and curation method.
>     - **Frontier models:** We include frontier models like OpenAI o4-mini, Qwen3, Qwen2.5, Llama3.1, not as direct competitors, but as crucial reference points to situate our work against the absolute state-of-the-art, per your suggestion.
> 2. **New SOTA Performance with SFT+RL!** To directly address your concern for stronger comparisons and demonstrate Guru's full potential, we conducted a new SFT+RL experiment, with results in Table 2. We conduct SFT on AM-Think-Distill-1.4M[1] dataset (base Qwen2.5-32b) and then perform RL with Guru dataset to further enhance the reasoning. Notably, our model achieves average performance **very close to frontier close reasoning model OpenAI o4-mini (55.7 vs. 56.2), outperforming frontier open reasoning models Qwen3-8B and QwQ-32B on AIME24** (we haven't finished QwQ-32B eval on all datasets due to time). Crucially, this improvement is not only from SFT; the RL phase with Guru was vital, boosting the AIME24 score from 78.0% (post-SFT) to 84.1%. While not the paper's primary focus, this result underscores Guru's effectiveness.
>
> ### Table 1: Performance of more general-domain open models baselines vs. Guru
>
> | Model | Open Data | w/ SFT? | AIME | LiveCodeBench | GPQA | ARC-AGI | HiTab | CodeIO | Avg. |
> | --- | --- | --- | --- | --- | --- | --- | --- | --- | --- |
> | Qwen2.5-7B-base | - | - | 4.2 | 8.2 | 32.4 | 0.6 | 41.1 | 6.4 | 15.5 |
> | Qwen2.5-7B-Instruct | - | yes | 11.7 | 12.6 | 40.3 | 1.9 | 54.8 | 8.5 | 21.6 |
> | Llama3.1-8B-Instruct | - | yes | 5.9 | 9.1 | **60.2** | 4.7 | 38.0 | 9.1 | 21.2 |
> | General Reasoner 7B | yes | - | 17.1 | 8.5 | 38.6 | 0.8 | 54.4 | 7.1 | 21.1 |
> | Qwen2.5-32B-base | - | - | 3.4 | 16.8 | 34.3 | 4.5 | 63.2 | 7.8 | 21.7 |
> | Qwen2.5-32B-Instruct | - | yes | 17.1 | 25.1 | 46.3 | 8.8 | 73.3 | 10.4 | 30.2 |
> | **Guru-7B (Ours)** | yes | - | 17.5 | 16.5 | 40.8 | 1.3 | 38.0 | 9.1 | 20.5 |
> | **Guru-32B (Ours)** | yes | - | **34.9** | **29.3** | 50.6 | **7.6** | **82.0** | **12.6** | **36.2** |
>
> ### Table 2: Performance of frontier reasoning models vs. Guru w/ SFT
>
> | Model | Open Data | Open Weight | w/ SFT? | AIME | LiveCodeBench | GPQA | ARC-AGI | HiTab | CodeIO | Avg. |
> | --- | --- | --- | --- | --- | --- | --- | --- | --- | --- | --- |
> | Qwen3-8B | - | yes | yes | 71.8 | 53.5 | 60.2 | 4.7 | 70.5 | 19.6 | 46.7 |
> | QwQ-32B | - | yes | yes | 79.5 | 62.7 | - | - | - | - | - |
> | OpenAI o4-mini | - | - | yes | 79.3 | **66.2** | **75.3** | **22.9** | 64.3 | **29.4** | **56.2** |
> | **Guru-32B w/ SFT (Ours)** | yes | yes | yes | **84.1** | 60.9 | 68.4 | 15.0 | **78.0** | 27.5 | 55.7 |
>
> ---
>
> > **W2**: The results in Table 2 (best validation accuracy within 200 RL steps) show that harder datasets can accelerate training, but not necessarily improve final performance. This suggests that simple queries contribute less to learning. However, it remains unclear whether the performance has already stabilized within 200 steps. There could still be improvements happening after that point, which are not shown in the table. I think stable results/performance is required.
> >
>
> Thank you for raising this important point about convergence. We confirm that for the ablation study in question, performance does indeed stabilize within the 200 RL steps. We provide two pieces of evidence:
>
> 1. **Reward curve plateau:** As shown in **Table 3**, the training reward for both configurations clearly plateaus after approximately 150 steps, indicating that the models have converged.
> 2. **Sufficient gradient updates:** This rapid convergence is expected given the total training volume. With a batch size of 512 and a mini-batch size of 32, each RL step involves 16 gradient updates. Therefore, the 200-step training run consists of a total of **3,200 gradient updates**, a substantial amount for a single-domain fine-tuning task.
>
> We hope this evidence clarifies that the 200-step mark is sufficient for observing stable final performance in this experiment.
>
> ### Table 3: Averaged Sample Reward in One Training Batch (batchsize=512)
>
> | Filtering Method | Step 0 | Step 50 | Step 100 | Step 150 | Step 180 | Step 190 | Step 200 | Step 210 | Step 220 |
> | --- | --- | --- | --- | --- | --- | --- | --- | --- | --- |
> | Difficulty filtering | 6.3 | 23.5 | 26.7 | 34.2 | 32.3 | 36.1 | 32.5 | 34.5 | 36.8 |
> | Non-difficulty filtering| 9.0 | 33.2 | 36.0 | 41.4 | 43.7 | 40.0 | 42.5 | 43.1 | 42.4 |
>
> ---
>
> >W3 Base Model Choice. Qwen is proven to work well for the RL method (magic), but it would be better to test at least one other open-source model besides Qwen. This would help demonstrate that good performance obtained from RL with your data is not dependent on Qwen’s specific behavior and can generalize across LLMs.
> >
>
> We sincerely thank the reviewer for raising this critical point. To verify that Guru's benefits generalize beyond the Qwen architecture, we have conducted the requested experiment. We replicated our cross-domain RL setup from Section 3 on Llama-3.1-8B-Instruct using the same Guru18k dataset.
>
> ### Table 4: Cross-domain transfer with Llama-3.1-8B-Instruct
>
> |Training Data             |Math |Coding|Science|Logic|Simulation|Table |
> |--------------------------|-----|-------|-------|-----|----------|------|
> |Base (Llama-3.1-8B-instruct)         |13.1 |38.9   |19.0   |6.6  |6.0       |25.5  |
> |+Math                     |25.7 |43.6   |25.5   |16.6 |10.0      |42.0  |
> |+Coding                   |23.9 |**52.6**   |26.0   |15.1 |11.0      |37.0  |
> |+Science                  |20.4 |45.5   |25.5   |13.0 |8.5       |34.5  |
> |+Logic                    |23.4 |44.9   |25.5   |**40.6** |9.0   |40.0  |
> |+Simulation               |26.9 |46.3   |29.0   |17.7 |15.0      |41.3  |
> |+Table                    |26.7 |44.5   |27.5   |15.9 |11.0      |62.2|
> |+Guru18k                  |**28.9**|51.6|**29.0**|39.8 |**16.0**|**62.5**|
>
>
> **The results confirm that our central findings generalize across architectures.** As Table 4 shows, training on the diverse Guru18k dataset achieves substantial gains over the base model and yields performance comparable to or better than any single-domain fine-tuning. This directly demonstrates that the cross-domain reasoning improvements facilitated by Guru are not dependent on a specific model family.
>
> We also have two brief observations from this new experiment:
>
> 1. **Base vs. Instruct models:** We initially found that the Llama-3.1-8B-base model could not learn effectively from RL due to its limited instruction-following ability. Switching to the aligned Llama-3.1-8B-Instruct model was necessary for successful training, highlighting the importance of a capable base model for RL.
> 2. **Data difficulty interaction:** We observed that Llama-3.1-8B-Instruct struggled with our high-difficulty math subset when trained alone. This suggests an interesting interaction between a model's inherent capabilities and data difficulty, where weaker models may require a gentler curriculum—a valuable insight that aligns with prior work like SimpleRL [2].
>
> We would like to thank you again for your constructive and highly valuable feedback. In response to your suggestions, we have conducted extensive new experiments: we added 8 new baselines, demonstrated state-of-the-art performance with SFT+RL, confirmed our training convergence, and validated our core findings on the Llama3 architecture. We will include these findings in our next paper version.
>
> We hope that these substantial additions have accomplished exactly that, "wish the paper could be stronger and give me more confidence in using these data.”, strengthening the paper and validating the general utility of Guru. We hope you'll agree and that our updated submission merits your full support.
>
> **References:**
> + [1]: 1.4 Million Open-Source Distilled Reasoning Dataset to Empower Large Language Model Training
> + [2]: SimpleRL-Zoo: Investigating and Taming Zero Reinforcement Learning for Open Base Models in the Wild

---

> > ### Comment · Reviewer_5khS · 2025-08-04
> >
> > Thanks for so many new experiments!!!! I appreciate it a lot.
> >
> > For W2, I am not convinced by results in STEP 220 to show that it is stable. Could you please LMK how many examples we have used in 200 steps?
> >
> > For W1 and W3, I think it looks good.

---

> > > ### Author Response · Authors · 2025-08-04
> > > **Further Response to W2**
> > >
> > > Glad to hear that our experiments addressed W1 and W3! We'd like to provide additional details regarding W2.
> > >
> > > First, in our setup, each batch contains 512 samples, with 16 rollouts per sample. Over 220 steps, this amounts to approximately 113K samples and 1.8 million rollouts. For reference, we collected some typical 7B-model RL experimental configs:
> > >
> > > * **SimpleRL** \[1] conducted \~120 steps (estimated from its RL plot) with batch size 1024.
> > > * **Spurious Rewards** \[2] ran 300 steps with batch size 128.
> > > * **OpenReasonerZero** \[3] used \~700 steps (estimated from its RL plot; batch size not disclosed in the paper), including a late-stage annealing phase.
> > >
> > > These comparisons suggest our training scale is consistent with prior work. As this experiment focuses on data ablation rather than maximizing performance, we defer longer training for peak scores to future work.
> > >
> > > Second, we observe that not only do the training rewards plateau around step 200, but most downstream evaluation metrics also saturate or slightly degrade beyond this point. Below are the accuracies on benchmarks around step 200 for the Math Filtering run:
> > >
> > > | Benchmark         | Step 150 | 180  | 200  | 220  |
> > > | ----------------- | -------- | ---- | ---- | ---- |
> > > | **MATH500**       | 76.0     | 75.8 | 77.0 | 75.4 |
> > > | **AMC**           | 54.5     | 54.5 | 54.5 | 53.3 |
> > > | **AIME**          | 17.9     | 17.9 | 14.5 | 13.3 |
> > > | **HumanEval**     | 56.7     | 51.8 | 56.1 | 46.9 |
> > > | **LiveCodeBench** | 6.8      | 10.7 | 8.2  | 5.7  |
> > > | **HiTab**         | 47.5     | 49.5 | 42.5 | 47.0 |
> > > | **MultiHierr**    | 33.0     | 30.0 | 35.0 | 34.0 |
> > >
> > > We hope this clarification fully resolves the concern and provides a clearer picture of the stability of our training. Thank you again for your thoughtful feedback—we truly appreciate your time and support. Given there is some divergence in reviewer scores, a clear and consolidated positive assessment would be crucial for a fair final decision. We trust that our provided evidence solidifies the case for acceptance.
> > >
> > > ---
> > >
> > > **References**
> > >
> > > + [1] *SimpleRL-Zoo: Investigating and Taming Zero Reinforcement Learning for Open Base Models in the Wild*
> > > + [2] *Spurious Rewards: Rethinking Training Signals in RLVR*
> > > + [3] *Open Reasoner Zero: An Open Source Approach to Scaling Up Reinforcement Learning on the Base Model*

---

> > > > ### Comment · Reviewer_5khS · 2025-08-06
> > > >
> > > > Thanks, will increase to 5.

---

### Official Review · Reviewer_xCcn · 2025-07-02

**Rating:** 3
**Confidence:** 4

**Summary:**

This paper introduces a 92K-sample dataset for RL for LLM Reasoning, covering six domains: Math, Code, Science, Logic, Simulation, and Tabular. The authors employed a specially designed pipeline for data collection and processing. The paper also investigates the transfer learning capabilities across different tasks and trains two models, achieving performance improvements.

**Dataset Code Accessibility:**

Yes

**Dataset Code Comments:**

N/A

**Ethical Considerations:**

No, there are no or only very minor ethics concerns

**Final Justification:**

The rebuttal addresses some of my concerns.

**However**, some concerns still remain. Specifically, the paper’s reported average improvement seems to be driven primarily by certain tasks that other baseline models may not include (see Table 1 in the rebuttal), which could lead to an unfair comparison of the “average” results.

Although the authors performed more fine-grained hyperparameter tuning in their response to W3, this issue persists (Table 4 in the rebuttal). For example, on mathematical tasks common to both the ORZ and GURU datasets, Guru-32B (clip-high=0.28) still trails ORZ-32B on AIME24 significantly. Furthermore, the authors have not reported the post-tuning results for MATH500 and LiveBench, which I identified as weaknesses in my original review.

Considering this, I believe the authors need to conduct fairer experiments and comparisons, which is important.

**Limitations Weaknesses:**

- The paper claims to cover "Diverse domains," but six domains may not be sufficient to be considered truly diverse.

- The selection of baselines for comparison is insufficient. The paper primarily compares its models against those trained on math or code reasoning tasks, with only a few models trained on general-purpose tasks. Therefore, the paper should include more and stronger models (both open-source and closed-source) to better demonstrate the advantages brought by the GURU dataset.

- The performance gains from the GURU dataset are most significant in domains where other baselines/papers have not been trained, such as the Logic domain. In contrast, in commonly benchmarked domains like MATH500 and LiveBench, the improvements are marginal or even weaker.

- The data is primarily sourced from existing public datasets on the internet, which somewhat diminishes the paper's contribution.

**Strengths Contributions:**

- The constructed dataset spans six domains and holds intrinsic value.

- The paper's investigation into the transfer relationships between different tasks offers valuable insights.

---

> ### Author Rebuttal · Authors · 2025-07-31
>
> We thank the reviewer for their constructive feedback and for recognizing the value of our dataset and analysis. We understand the concerns that led to the initial rating, and in this rebuttal, we have worked to address each point directly with new evidence and experiments.
>
> ---
> > W1: The paper claims to cover "Diverse domains," but six domains may not be sufficient to be considered truly diverse.
> >
>
> Thank you for raising this important point. We agree that a simple domain count is an incomplete metric for diversity. Our claim is rooted in two deeper dimensions: 1) the rich variety of reasoning skills and task formats, and 2) the significant, rigorous curation effort that makes Guru a unique resource.
>
> 1. **Diversity lies in reasoning skills and task formats** Rather than treating domains as monolithic blocks, we designed Guru to cover a wide spectrum of reasoning challenges. We address the in-domain task diversity and unique reasoning skill involved.
>     - **Logic Reasoning**: Includes Zebra puzzles (multi-attribute constraints), Ordering puzzles (global ordering from local relations), Graph puzzles (symbolic reasoning on textual graphs), and BARC puzzles (inspired by ARC-AGI, requiring abstract rule induction).
>     - **Simulation**: Employs Code IO tasks requiring mental execution and reverse causal reasoning of complex code.
>     - **Hierarchical Tables**: Uses complex tables with nested headers, demanding multi-hop inference beyond flat table reasoning.
>     - **Advanced STEM**: Focuses on university- and PhD-level tasks, excluding simpler formats (multiple-choice/boolean) to emphasize genuine scientific reasoning.
>     - **Math & Code**: Achieves breadth by aggregating multiple high-quality datasets (Appendix Figures 1 & 2), spanning algebraic, geometric, dynamic programming, and recursive problem-solving.
> 2. **(Table 1) Contribution to the community** Prior to our work, public RL datasets for reasoning were heavily skewed towards Math and Code. Guru is, to our knowledge, the first public multi-domain RL dataset with verified rewards by the time of submission. Please check Table 1 for details.
>
> 3. Finally, we want to address **Guru is built with substantial human engineering and GPU compute:** Each domain underwent a complete pipeline including data sourcing, reward design, deduplication, domain-specific filtering, and difficulty-based filtering. The first four stages are entirely domain-specific, requiring contributors to implement tailored filtering and reward functions for each sub-dataset, followed by trial RL experiments to verify meaningful improvements. This resulted in **24 data processing files and 14 reward function implementations (tested via small-scale 3B model RL before inclusion)**. Difficulty-filtering alone required approximately **7800 GPU hours** on NVIDIA H200s, inferencing 210k examples with Qwen2.5-7B-Instruct and Qwen3-30B-MoE-A3B (16 responses per sample) using vLLM.
>
>
> ### Table1: Comparison of open RL reasoning datasets
> | | |**Domain**| | | | | |**Data Processing**| |
> |-|-|-|-|-|-|-|-|-|-|
> |**RL Dataset**|**Dataset size**|**Math**|**Code**|**STEM**|**Logic**|**Sim.**|**Table**|**(Model-Based) Difficulty Filtering**|**Reward**|
> |**DeepScaler**|40.3k|yes|-|-|-|-|-|-|rule|
> |**BigMath**|251k|yes|-|-|-|-|-|-|rule|
> |**OpenReasonerZero**|57k|yes|-|-|-|-|-|yes|rule|
> |**OpenReasonerOne**|119k|yes|-|-|-|-|-|yes|rule|
> |**DAPO**|17k|yes|-|-|-|-|-|-|rule|
> |**KodCode**|487k|-|yes|-|-|-|-|yes|execution|
> |**DeepCoder**|24k|-|yes|-|-|-|-|-|execution|
> |**GeneralReasoner**|120k|-|-|yes|-|-|-|yes|llm-as-judge|
> |**ReasoningGym**|>100 task generators|-|-|-|yes|-|-|-|rule|
> |**Guru (Ours)**|92k|yes|yes|yes|yes|yes|yes|yes (weak & strong models)|rule & execution & llm-as-judge|
>
>
>
> ---
>
> > W2: The selection of baselines for comparison is insufficient.
> >
>
> Thank you for your constructive feedback regarding the selection of baselines. We recognize the importance of clarifying the role of the Guru dataset and our baseline selection. Besides, we run additional SFT+RL with Guru to achieve **on-par performance with frontier closed OpenAI o4-mini-high performance.**
> - **Guru dataset focuses on open data for RL.** Unlike frontier open-weight reasoning models like Qwen3, and close ones like OpenAI-o4-mini, our Guru models do not undergo SFT before RL. Data used in Qwen and OpenAI series are publicly unavailable. SFT significantly boosts performance and is different from our focused RL phase in this paper. Nonetheless, we also demonstrate the quality of Guru data with RL after SFT below.
> - **(Table 2 & 3) We add 8 more baseline model evaluations.** Due to time limits we evaluated on one representative task per domain. Our core comparison remains with General Reasoner, which, like our model, was trained on open general reasoning data. As shown in the paper, Guru-7B significantly outperforms General-Reasoner-7B across most tasks.
> - **(Table 3) Guru-32B with SFT is on par with the frontier close OpenAI reasoning model o4-mini.** We conduct SFT on AM-Think-Distill-1.4M[1] dataset (base Qwen2.5-32b) and then perform RL with Guru dataset to further enhance the reasoning. Notably, our model achieves average performance very close to frontier close reasoning model OpenAI o4-mini (55.7 vs. 56.2), outperforming frontier open reasoning models Qwen3-8B and QwQ-32B on AIME24 (we haven't finished QwQ-32B eval on all datasets due to time). Crucially, this improvement is not only from SFT; the RL phase with Guru was vital, boosting the AIME24 score from 78.0% (post-SFT) to 84.1%. While not the paper's primary focus, this result underscores Guru's effectiveness.
> - **Analytical insights**: Guru shows that findings from math/code domains do not generalize to other reasoning areas, a key insight enabled by Guru's diversity (Sec. 3)
>
> ### Table 2: Performance of more general-domain open models baselines vs. Guru
> |Model|Open Data|w/ SFT?|AIME|LCB|GPQA|ARC-AGI|HiTab|CodeIO|Avg.|
> |-|-|-|-|-|-|-|-|-|-|
> |Qwen2.5-7B-base|-|-|4.2|8.2|32.4|0.6|41.1|6.4|15.5|
> |Qwen2.5-7B-Instruct|-|yes|11.7|12.6|40.3|1.9|54.8|8.5|21.6|
> |Llama3.1-8B-Instruct|-|yes|5.9|9.1|**60.2**|4.7|38.0|9.1|21.2|
> |General Reasoner 7B|yes|-|17.1|8.5|38.6|0.8|54.4|7.1|21.1|
> |Qwen2.5-32B-base|-|-|3.4|16.8|34.3|4.5|63.2|7.8|21.7|
> |Qwen2.5-32B-Instruct|-|yes|17.1|25.1|46.3|8.8|73.3|10.4|30.2|
> |**Guru-7B (Ours)**|yes|-|17.5|16.5|40.8|1.3|38.0|9.1|20.5|
> |**Guru-32B (Ours)**|yes|-|**34.9**|**29.3**|50.6|**7.6**|**82.0**|**12.6**|**36.2**|
>
> ### Table 3: Performance of frontier reasoning models vs. Guru w/ SFT
> |Model|Open Data|Open Weight|w/ SFT?|AIME|LCB|GPQA|ARC-AGI|HiTab|CodeIO|Avg.|
> |-|-|-|-|-|-|-|-|-|-|-|
> |Qwen3-8B|-|yes|yes|71.8|53.5|60.2|4.7|70.5|19.6|46.7|
> |QwQ-32B|-|yes|yes|79.5|62.7|-|-|-|-|-|
> |OpenAI o4-mini|-|-|yes|79.3|**66.2**|**75.3**|**22.9**|64.3|**29.4**|**56.2**|
> |**Guru-32B w/ SFT (Ours)**|yes|yes|yes|**84.1**|60.9|68.4|15.0|**78.0**|27.5|55.7|
>
> ---
>
> > W3: The performance gains from the Guru dataset are not strong enough on certain tasks.
> >
>
> Thank you for this insightful point. We find this is also closely related to our response to W2, where we show Guru-32B and Guru-32B (w/ SFT) achieve SOTA performance on general domains.
>
> We also notice that final performance is highly sensitive to training hyperparameters beyond the dataset itself. For instance, SimpleRL sweeps system prompts for math; ORZ uses a final annealing stage that boosts its AIME score by ~10%. To illustrate the sensitivity to tuning, we increased the high clip ratio of GRPO algorithm from 0.2 (used in our paper) to 0.28, encouraging greater exploration, and observed consistent improvements:
>
> ### Table 4: Guru-32B performance with higher clip ratio
>
> |Config|AIME|LCB|GPQA|ARC-AGI|HiTab|CodeIO|Avg.|
> |-|-|-|-|-|-|-|-|
> |**Guru-32B**|34.9|**29.3**|50.6|7.6|82.0|12.6|36.2|
> |**Guru-32B (clip-high=0.28)**|**38.9**|28.5|50.6|**7.7**|**82.2**|**19.2**|**37.9**|
>
> ---
>
> > W4: The data is primarily sourced from existing public datasets.
> >
> Thank you for your valuable feedback. We respectfully argue that our work's contribution lies in its rigorous and resource-intensive curation, which is fully aligned with the scope of the NeurIPS D&B track.
>
> 1. The NeurIPS D&B track explicitly values datasets created via "careful and thoughtful" curation of existing public sources. Precedents like Tulu[2] and RedPajama[3] confirm that the primary contribution can be the curation pipeline and analysis, rather than sourcing novel data alone.
> 2. Our contribution is precisely this intensive curation. As detailed in our response in W1, the Guru pipeline involved substantial human engineering and GPU compute for multi-domain data curation. Below in Table 5, we demonstrate how the **difficulty filtering directly translates into RL performance gains**, validating our method.
> 3. Beyond repurposing existing datasets, we synthesized four new, scalable Logic datasets from scratch (Zebra, Order, Graph, and BARC puzzles), adding originality to our dataset.
>
> ### Table 5: Ablation study on difficulty filtering
> | Base Model | Config | Math | Code | STEM | Logic | Table | Sim. |
> | --- | --- | --- | --- | --- | --- | --- | --- |
> | **Qwen2.5-7B-Base** | Guru (ours) | **51.1** | **59.5** | **37.7** | **49.4** | **58.5** | **14.5** |
> |  | └─ w/o difficulty filtering | 50.2 | 59.3 | 37.0 | 48.8 | 56.3 | 12.5 |
> | **Qwen2.5-32B-Base** | Guru (ours) | **62.9** | 68.7 | **49.8** | **64.3** | **72.3** | **20.5** |
> |  | └─ w/o difficulty filtering | 59.9 | 68.7 | 49.6 | 60.0 | 71.8 | 20.0 |
>
> We hope this new evidence addresses the concerns you raised. We hope you will reconsider your evaluation in light of these substantial updates. Thank you again for your time and valuable feedback!
>
> **References**:
> - [1]: 1.4 Million Open-Source Distilled Reasoning Dataset to Empower Large Language Model Training
> - [2]: How Far Can Camels Go? Exploring the State of Instruction Tuning on Open Resources
> - [3]: RedPajama: an Open Dataset for Training Large Language Models

---

> > ### Comment · Reviewer_xCcn · 2025-08-08
> >
> > Thank you for the response. It addresses some of my concerns, so I will raise my score by 1.
> >
> > **However, some main concerns still remain. ** Specifically, the paper’s reported average improvement seems to be driven primarily by certain tasks that other baseline models may not include (see Table 1 in the rebuttal), which could lead to an unfair comparison of the “average” results.
> >
> > Although the authors performed more fine-grained hyperparameter tuning in their response to W3, this issue persists (Table 4 in the rebuttal). For example, on mathematical tasks common to both the ORZ and GURU datasets, Guru-32B (clip-high=0.28) still trails ORZ-32B on AIME24 significantly. **Furthermore, the authors have not reported the post-tuning results for MATH500 and LiveBench, which I identified as weaknesses in my original review.**
> >
> > Considering this, I believe the authors need to conduct fairer experiments and comparisons to make it more solid.

---

> > > ### Author Response · Authors · 2025-08-08
> > > **Reponse by Authors**
> > >
> > > Thank you for the detailed follow-up. We appreciate the opportunity to offer a final set of clarifications based on your comments.
> > >
> > > 1. **On Guru Dataset Position and the ORZ Comparison:**
> > > + As pointed out in our rebuttal (W2), the most direct baseline for our "general-reasoning" claims is General-Reasoner-7B (the only fully public effort of general LLM reasoning with RL to our knowledge upon submission), which **Guru-7B significantly surpasses across almost all domains.**
> > > + We included math-specialist models like ORZ as baselines to highlight our core motivation: to show how the open community's focus on "hill-climbing" math benchmarks is insufficient for creating general reasoners that can rival models like o4-mini. Also, ORZ is heavily optimized for math, using customized techniques like a final annealing stage (Appendix B) and exhaustive PPO GAE hyperparameter tuning (Section 2.2) to maximize AIME scores. As a reference, before this final annealing, **ORZ’s AIME score was ~41% (estimated from their official plot), which is very close to our Guru model's performance of 38.9% (rebuttal Table 4)**.
> > >
> > > 2. **On the Significance of the SFT+RL Experiment:** Our additional experiments show that with SFT followed by RL on Guru, our model can achieve performance **on par with OpenAI’s o4-mini and surpass other frontier (arguably the best) open models like Qwen3 and QwQ**. We chose this more resource-intensive but realistic scenario SFT+RL because we believe it provides** the strongest possible validation of our dataset’s quality across all domains**, including math. We feel this is a more significant and compelling result than narrowly competing with a pure-RL specialist like ORZ.
> > >
> > > 3. **Results for MATH500 and LiveBench:** We have now evaluated our model on MATH500 and LiveBench as requested in the table below. As mentioned in the first rebuttal response, we didn't put them in the rebuttal Table 4 because we only picked one representative task per domain due to time constraints. With our clip-high=0.28 tuning, our model **improves on MATH500 and consistently outperforms ORZ on LiveBench**.
> > >
> > > | Model | MATH500 | LiveBench |
> > > | --- | -- | :---: |
> > > | Qwen2.5-32B-Base | 49.6 | 22.7 |
> > > | ORZ-32B | **89.8** | 29.8 |
> > > | **Guru-32B (ours, clip-high=0.2)** | 84.0 | 34.3 |
> > > | **Guru-32B (ours, clip-high=0.28)** | 86.8 | **35.4** |
> > >
> > >
> > > To summarize our final points in response to the remaining concern about math performance compared to ORZ:
> > > + We re-clarify that Guru's primary goal is to advocate for general reasoning, and our Guru-7B model largely beats the most comparable baseline, General-Reasoner-7B.
> > > + We show that our Guru-32B model is comparable to ORZ-32B (38.9% vs. ~41% on AIME) before its math-specific final annealing stage.
> > > + We demonstrate that in a more realistic SFT+RL scenario, our dataset helps achieve true SOTA general-reasoning performance.
> > > + We have provided the requested MATH500 and LiveBench results to resolve any misunderstandings.
> > >
> > > We hope this comprehensive evidence could help justify the quality and contribution of our Guru dataset. Thank you for helping us sharpen the paper's message!

---

> ### Author Response · Authors · 2025-08-06
> **Follow-up regarding our rebuttal**
>
> Dear Reviewer xCcn,
>
> Thank you again for your time and the detailed feedback on our paper.
>
> Following up on your comments, we posted our detailed rebuttal with substantial new experiments and clarifications specifically designed to address the points you raised. For your convenience, here is a brief summary:
>
> 1. **On Insufficient Baselines & Performance**: In response to the reviewer's concerns, we added 8 new baselines. More importantly, we ran a new SFT+RL experiment using our Guru dataset. As shown in our rebuttal (Table 3), this elevates a 32B model to achieve performance **on par with OpenAI's frontier o4-mini model** on a diverse set of reasoning tasks—a result we believe is of significant interest to the broader NeurIPS community.
>
> 2. **On Contribution (Data Sourcing)**: The reviewer questioned our contribution since we built on public data. We clarified that our work's contribution, a rigorous, resource-intensive curation pipeline **(7800+ GPU hours for difficulty filtering alone)**, is explicitly aligned with the scope of the NeurIPS Datasets & Benchmarks track, citing precedents like Tulu and RedPajama. We also noted that we synthesized four novel logic puzzle datasets from scratch.
>
> 3. **On Domain Diversity**: Regarding domain diversity, we clarified that the diversity not only lies in the domain but also the variety of reasoning skills that our dataset introduces to the public RL landscape. Our comparison table shows that Guru is **the first public RL for LLM reasoning dataset** to offer this breadth.
>
> We are very keen to hear your thoughts on these updates. As the discussion period concludes soon, we just wanted to politely follow up to ensure you've had a chance to see our response.
>
> Thank you again for your valuable time and consideration.

---

### Official Review · Reviewer_jPcD · 2025-07-03

**Rating:** 5
**Confidence:** 3

**Summary:**

The paper proposes GURU, a multi-domain dataset for training language models with reinforcement learning. Unlike existing datasets, which focus on single domains and primarily target math or code, this dataset covers Math, Code, Science, Logic, Simulation, and Tabular analysis. Using this proposed dataset, the paper provides a thorough analysis of the cross-domain transferability of existing language models. In addition, the paper proposes two new models, GURU-7B and GURU-32B, that were trained purely with RL on GURU and demonstrate significant improvements over leading open RL reasoning model baselines.

**Additional Feedback:**

Overall, the work is highly impactful and I don't have any further suggestions.

**Dataset Code Accessibility:**

Yes

**Dataset Code Comments:**

The dataset is available on Hugging Face.

**Ethical Considerations:**

No, there are no or only very minor ethics concerns

**Final Justification:**

Overall, I think this paper is solid - it provides both a benchmark and a useful cross-domain reasoning transfer analysis. Thus, I recommend accepting the paper.

**Limitations Weaknesses:**

Limitations:

The paper proposes a technique for data deduplication and two filtering techniques (heuristic filtering and difficulty filtering) to improve dataset quality. However, except for Table 2 (which shows ablation of difficulty filtering on math data), I couldn't find an ablation study that shows these techniques improve data quality.

Please address the following questions:

1) Please provide ablation studies demonstrating the effectiveness of the proposed filtering techniques.

2) In line 168 (Potentially noisy samples), the authors mention that they remove all samples that the strong model consistently fails, since they may be ambiguous or malformed. Why is this assumption valid? What if a stronger model could succeed on those examples?

3) Figure 1 (left) and Figure 3 show that performance on Logic, Simulation, and Tabular tasks slightly degrades when models are trained on the mixed dataset compared to single-domain datasets. While I expect that adding certain domains to the training data may not help improve others (for example, code may not help with tabular tasks), I don't expect additional data to degrade performance. Why does this occur?

4) In lines 115-117, when describing the deduplication process, the authors mention that in preliminary experiments, similarity-based metrics such as embedding distance, Jaccard similarity, and n-gram overlap are prone to false positives. Could you explain why and provide more examples?

**Strengths Contributions:**

* The GURU dataset tackles an important problem of cross-domain generalization of language models trained with reinforcement learning.

* The authors propose two unique filtering techniques that ensure robust and accurate reward signals.

* The paper is well-written and easy to follow.

* The paper provides an extensive analysis of cross-domain reasoning transfer that emphasizes the importance of training language models on diverse, multi-domain datasets.

* The paper presents two general reasoning models that were trained with RL on the GURU dataset and achieve state-of-the-art results.

---

> ### Author Rebuttal · Authors · 2025-07-31
>
> We sincerely thank the reviewer for their exceptionally thorough and encouraging review. We are particularly grateful that they recognized the core contributions of our work, from the dataset itself to the cross-domain analysis. The questions raised are all highly insightful and give us a valuable opportunity to elaborate on key methodological details. We address each point below, and we believe these clarifications will make the final version of our paper stronger.
>
> ---
> >W1: Please provide ablation studies demonstrating the effectiveness of the proposed filtering techniques.
> >
> Thanks for pointing that out. We conducted an additional study during the rebuttal phase across all six domains using both 7B and 32B base models. Specifically, we compare our full Guru data pipeline against a variant that removes difficulty filtering while keeping all other components fixed. We construct a 20k subset for each configuration and run RL for 200 steps with batch size 512. Evaluation is done on our online suite, where we report the best validation accuracy for each task:
>
> | Base Model  | Config   | Math | Code | STEM | Logic | Table | Simulation | Avg.   |
> | -- | -- | -- | -- | -- | --- | --- | -- | -- |
> | **Qwen2.5-7B-Base** | Guru (ours)                 | 51.1 | 59.5 | 37.7 | 49.4  | 58.5  |   14.5     | 45.12 |
> |                    | └─ w/o difficulty filtering | 50.2 | 59.3 | 37.0 | 48.8  | 56.3  |   12.5     | 44.02 |
> | **Qwen2.5-32B-Base**| Guru (ours)                 | 62.9 | 68.7 | 49.8 | 64.3  | 72.3  |   20.5     | 56.42 |
> |                    | └─ w/o difficulty filtering | 59.9 | 68.7 | 49.6 | 60.0  | 71.8  |   20.0     | 55.00 |
>
> We observe that difficulty filtering generally contributes to performance gains across different model sizes. On average, it improves performance by +1.1% for the 7B model (45.12 → 44.02) and +1.4% for the 32B model (56.42 → 55.00). The improvements are especially notable in domains like Math (+3.0%) and Logic (+4.3%) for the 32B model. This suggests that focusing RL training on more challenging and stable examples, as selected by our difficulty-aware filtering, is broadly beneficial.
>
> We hope this expanded ablation and clarification on the importance of the difficulty filtering.
>
> ---
> >W2: In line 168 (Potentially noisy samples), the authors mention that they remove all samples that the strong model consistently fails, since they may be ambiguous or malformed. Why is this assumption valid? What if a stronger model could succeed on those examples?
> >
> Thank you for this excellent question about our data filtering strategy. Our rationale was based on the capabilities of the models used in this study.
>
> At the time of our experiments, we used Qwen3-30B-MoE-A3B as our "strong model" for filtering. We reasoned that samples that consistently failed were likely either (1) malformed or noisy, or (2) intractably difficult for our base models to learn from effectively via RL.
>
> However, you raise a crucial point: this assumption is context-dependent and may not hold for even stronger future models. We agree that indiscriminately removing all such samples could discard valuable, high-difficulty data.
>
> Indeed, we have already encountered this scenario in our ongoing work. When applying RL to stronger, SFT-tuned base models, we no longer remove all zero-pass-rate samples. Instead, we now use a more nuanced filtering approach (e.g., maintaining a "mirrored-J" difficulty distribution) to preserve these challenging examples.
>
> We will update the manuscript to add a discussion of this important nuance. Thank you again for prompting this valuable clarification.
>
> ---
> >W3: Figure 1 (left) and Figure 3 show that performance on Logic, Simulation, and Tabular tasks slightly degrades when models are trained on the mixed dataset compared to single-domain datasets. While I expect that adding certain domains to the training data may not help improve others (for example, code may not help with tabular tasks), I don't expect additional data to degrade performance. Why does this occur?
> >
>
> Thank you for raising this point. We assume the slight dips on Logic, Simulation and Tabular stem from a few interacting factors:
> + **Gradient conflict across tasks**. When optimizing for very different objectives in one step, their gradients can collide. Math-focused updates may pull parameters one way, while tabular updates pull another, resulting in a compromise that slightly under-serves the less dominant domains.
> + **Faster forgetting under sparse sampling**. In a six-domain mix, each domain appears in only about one-sixth of mini-batches. Logic and Simulation examples may “decay” between updates, whereas in single-domain training they benefit from continuous reinforcement [1].
> + **Inherent RL noise**. Policy gradient methods like PPO are inherently noisy. Small performance swings of 1-2 points are often within the standard variance of identical training runs, especially on complex, long-horizon reasoning tasks. The observed dips are minor and may be partially attributable to this stochasticity.
>
> Together, these small effects—gradient interference, forgetting, and RL variance—can add up to a few-point drop, even though overall capability and average reward continue to improve.
>
> ---
> >W4: In lines 115-117, when describing the deduplication process, the authors mention that in preliminary experiments, similarity-based metrics such as embedding distance, Jaccard similarity, and n-gram overlap are prone to false positives. Could you explain why and provide more examples?
> >
> We thank the reviewer for the question and fully understand the concern. In reasoning tasks, determining whether two samples are truly duplicated is indeed a nuanced and non-trivial challenge. In our preliminary experiments, we observed that some problems, despite having highly similar surface forms, like templated language, or overlapping entities, may differ significantly in their constraints and reasoning pathways, ultimately requiring entirely different solutions and strategies. To preserve such structurally similar but logically diverse high-quality examples and encourage the model to learn a wider range of reasoning paths, we opted for a deliberately conservative deduplication strategy to avoid mistakenly removing valuable data, (e.g., "is next to" vs. "is not next to").
>
> That said, we agree with the reviewer’s observation: this conservative approach can overlook certain near-duplicate examples. Through subsequent rounds of in-depth manual review, we identified such cases as well, and we believe this is a highly worthwhile direction for further investigation. In future iterations, we plan to explore more fine-grained, subdomain-aware deduplication strategies to better balance data quality, consistency, and coverage.
>
> **Reference**
> + [1] A Survey on Negative Transfer

---

### Official Review · Reviewer_qEgP · 2025-07-07

**Rating:** 5
**Confidence:** 4

**Summary:**

The paper presents GURU, a 92K-example, multi-domain dataset curated specifically for RL-based reasoning in LLMs. GURU spans six reasoning domains: Math, Code, Science, Logic, Simulation, and Tabular, each paired with domain-specific verifiers that provide automated, high-fidelity reward signals. The authors design a principled five-stage data curation pipeline and conduct extensive cross-domain transfer and difficulty ablation experiments. The authors release two RL-trained models (GURU-7B and GURU-32B) that demonstrate SoTA performance on a 17-task evaluation suite, outperforming previous open-source RL-trained models.

**Dataset Code Accessibility:**

Yes

**Ethical Considerations:**

No, there are no or only very minor ethics concerns

**Limitations Weaknesses:**

- The paper lacks ablations isolating the effect of different pipeline stages, e.g., how much performance gain is attributable to difficulty filtering vs. heuristic filtering. I would suggest the authors to include controlled experiments disabling individual pipeline stages or reward types to quantify their relative contributions.

- No SFT Baseline Comparisons: The study avoids comparing GURU-RL models to similarly sized models fine-tuned with SFT on the same data. While this is explained as intentional to isolate RL effects, it would still be valuable to understand how much RL adds over SFT.

- The paper does not provide confidence intervals or variance measures for its main experiments, limiting robustness of the performance claims. Even small-scale repeat experiments (e.g., on 7B with GURU-18K) could help substantiate the consistency of gains.

- The English- and STEM-heavy nature of the dataset may limit the generality of the trained models. I would suggest the authors to include dataset composition statistics by linguistic style, problem format, and cultural/geographic origin.

- Training GURU-32B with RL requires 16 Hopper nodes over multiple days. I would suggest to include scaling analysis for smaller models (e.g., 3B or 1.5B), or provide efficiency metrics (e.g., reward improvement per GPU-hour).

- While tasks like Zebra Puzzle and Logic Graphs suggest some temporal reasoning, the paper does not clearly evaluate long-horizon tasks. Please add or highlight benchmarks that explicitly test multi-step chain-of-thought or deductive planning.

**Strengths Contributions:**

- Substantial Dataset Innovation: GURU addresses a key gap in RL-for-reasoning research by moving beyond Math and Code to cover six diverse domains. The pipeline includes thoughtful steps like deduplication, heuristic filtering, and difficulty calibration using empirical model pass rates, enhancing reward reliability.

- Automatic, Domain-Specific Reward Functions: The paper categorizes and applies different types of verifiers (rule-based, execution-based, and model-based). The paper provides a nuanced understanding of domain challenges.

- Rigorous Cross-Domain Transfer Analysis: The authors conduct controlled experiments using GURU-18K and systematically analyze transferability, difficulty impacts, and response-length dynamics, offering rare insight into how reasoning generalizes.

- Strong Empirical Results: GURU-7B and GURU-32B outperform competitive baselines by 7–8 percentage points on average across 17 benchmarks spanning all six domains. The dataset, models, and code will be released, supporting broader accessibility in open-source RL reasoning research.

- The paper is clearly structured with informative visualizations (e.g., Figures 1, 3, 4) and extensive benchmark coverage. The ablations (e.g., Table 2) are insightful and well-motivated. As reasoning-centric LLMs dominate frontier research, a reproducible, diverse RL dataset is of high significance. GURU positions itself as a strong alternative to proprietary RL training corpora like those used for GPT-4 or Claude.

---

> ### Author Rebuttal · Authors · 2025-07-31
>
> We sincerely thank you for your thoughtful and encouraging review. We are especially grateful that our core goals for this project came across clearly, from the careful design of the diverse, multi-domain Guru dataset to the rigorous experimental validation and our commitment to open and reproducible research. Your recognition of our work as a meaningful step forward for the community is very motivating.
>
> We also appreciate your constructive suggestions for improvement, which we address in detail below.
>
> ---
> > W1: The paper lacks ablations isolating the effect of different pipeline stages.
>
> Thank you for the thoughtful suggestion. We agree that isolating the contributions of different pipeline stages is important for better understanding the impact of each component.
>
> In the main paper (Section 3.3), we conducted an ablation on difficulty filtering in the Math domain and observed consistent improvements, particularly on harder tasks like AMC (+6.3%) and AIME24 (+5.9%). To further address your concern and provide a more holistic analysis, we conducted an additional study during the rebuttal phase across all six domains using both 7B and 32B base models.
>
> Specifically, we compare our full Guru data pipeline against a variant that removes difficulty filtering while keeping all other components fixed. We construct a 20k subset for each configuration and run RL for 200 steps. Evaluation is done on our online suite, where we report the best validation accuracy for each task:
>
> |Base Model|Configuration|Math|Code|STEM|Logic|Table|Simulation|Avg.|
> |-|-|-|-|-|-|-|-|-|
> |**Qwen2.5-7B-Base**|**Guru (ours)**|51.1|59.5|37.7|49.4|58.5|14.5|45.12|
> | |└─ w/o difficulty filtering|50.2|59.3|37.0|48.8|56.3|12.5|44.02|
> |**Qwen2.5-32B-Base**|**Guru (ours)**|62.9|68.7|49.8|64.3|72.3|20.5|56.42|
> | |└─ w/o difficulty filtering|59.9|68.7|49.6|60.0|71.8|20.0|55.00|
>
>
> We observe that difficulty filtering generally contributes to performance gains across different model sizes. On average, it improves performance by +1.1% for the 7B model (45.12 → 44.02) and +1.4% for the 32B model (56.42 → 55.00). The improvements are especially notable in domains like Math (+3.0%) and Logic (+4.3%) for the 32B model. This suggests that **focusing RL training on more challenging and stable examples, as selected by our difficulty-aware filtering, is broadly beneficial.**
>
> The reward types are domain-specific by design. For Math and Code, where verification is relatively straightforward, we use rule-based verifiers—e.g., extracting final answers from boxed notation in Math or executing unit tests in Code. In contrast, for STEM domains where answers are often free-form and cannot be easily matched using string comparison or regex, we employ model-based verifiers (i.e., LLM-as-a-judge) to provide more reliable reward signals.
>
> We hope this expanded ablation and clarification provide the insight you were seeking into the relative importance of each pipeline stage.
>
> ---
> > W2: No SFT Baseline Comparisons
>
> Thank you for the suggestion. While our primary focus is on using Guru-92K for RL from base models, we agree that examining its utility for SFT-pretrained models is valuable.
>
> To that end, we conducted an additional experiment where we apply RL on top of Distilled-R1-Qwen-7B—an SFT model from DeepSeek.
>
> The results are summarized below:
>
> | Model             | Math | AMC  | AIME | HumanEval | MBPP | LCB  | Zebra | Ordering | Graph | ARC-AGI | Code IO | HiTab | MultiHiertt | GPQA | SuperGPQA | Avg   |
> | :--- | :--: | :--: | :--: | :-------: | :--: | :--: | :---: | :------: | :---: | :-----: | :-----: | :---: | :----------: | :--: | :--------: | :---: |
> | **Distilled-7B**      | 92.8 | 82.5 | 53.3 |   90.2    | 72.0 | 28.3 |  0.0  |   62.0   |  3.8  |   0.0   |  16.0   | 40.5  |    31.5      | 37.8 |    25.0    | 42.38 |
> | + **Guru-RL (ours)**  | 93.6 | 84.6 | 63.3 |   92.7    | 74.5 | 28.3 | 20.2  |   78.0   | 18.2  |   2.5   |  16.0   | 50.5  |    39.5      | 48.9 |    29.0    | 49.32 |
>
> We observe consistent improvements across domains after applying RL with Guru—even on a strong SFT baseline. This highlights that **Guru-92K is also effective post-SFT, not just from base models.**
>
> ---
> > W3: The paper does not provide confidence intervals or variance measures for its main experiments, limiting the robustness of the performance claims
>
> We agree that reporting confidence intervals or variance measures can strengthen the robustness of performance claims. While full-scale repeated training runs are prohibitively expensive for large models—especially for experiments like Guru-18K which involve seven separate RL runs—we have taken steps to address variance where feasible:
>
> Main Results (Table 3): For evaluation benchmarks with ≤500 examples (e.g., AIME24, ARC-AGI), we perform repeated inference and report the average score across 32 samples per instance. ARC-AGI is scored using Pass@8; all other tasks report Avg@k with k ∈ {1, 8, 32}, following the R1 inference setup (T=0.6, Top-p=0.95).
>
>
> Transfer Trend Analysis (Figures 1 & 3): Our main insights focus on qualitative trends rather than exact performance numbers. Specifically, we observe that domains with heavier pretraining exposure (e.g., Math, Code, STEM) show stronger transferability, while underrepresented domains (e.g., Logic, Simulation, Tabular) benefit more from within-domain training. These patterns are consistent and robust across configurations.
>
>
> We hope these practices and findings help convey the stability and generality of our conclusions, even in the absence of full repeated training runs.
>
> ---
> > W4: I would suggest the authors to include dataset composition statistics by linguistic style, problem format, and cultural/geographic origin.
>
> We agree that the current composition of Guru mainly focuses on English and STEM questions, and we see expanding beyond this scope as a valuable direction for future work:
> Multilingual coverage is important for improving accessibility and fairness. We plan to incorporate multilingual reasoning problems in future versions of Guru.
>
> Our emphasis on STEM domains is primarily driven by the need for verifiable reward signals, which are essential for scalable RL. These domains offer structured tasks where correctness can be automatically assessed via symbolic matching, execution-based tests, or model-based verification.
>
> We also recognize the value of curating datasets with diverse linguistic styles, problem formats, and cultural or geographic origins, especially for aligning LLMs on open-domain or real-world reasoning tasks. While such efforts are nontrivial at scale, we consider them promising directions for future development.
>
> We appreciate the suggestion and will reflect these considerations in the revised paper.
>
> ---
> > W5: I would suggest to include scaling analysis for smaller models (e.g., 3B or 1.5B), or provide efficiency metrics (e.g., reward improvement per GPU-hour).
>
> Thank you for the suggestion. We agree that reporting efficiency metrics is important for evaluating the practical utility of RL training. We trained Guru models over 2.5 days on Hopper GPUs:
>
> **Guru-7B** used 4 nodes (32 GPUs) → 1920 GPU-hours
>
> **Guru-32B** used 16 nodes (128 GPUs) → 7680 GPU-hours
>
> We observe the following improvements in average score (over 6 representative tasks):
> | Model            | AIME | LiveCodeBench | GPQA | ARC-AGI | HiTab | CodeIO | Avg.  | GPU-hours | ∆ Score vs Base | Points per GPU-hour |
> |------------------|:----:|:-------------:|:----:|:-------:|:-----:|:------:|:-----:|:----------:|:----------------:|:---------------------:|
> | **Qwen2.5-7B (Base)** | 4.2  | 8.2           | 32.4 | 0.6     | 41.1  | 6.4    | 15.5  |     –      |        –         |          –            |
> | **Guru-7B (Ours)**    | 17.5 | 16.5          | 40.8 | 1.3     | 38.0  | 9.1    | 20.5  |   1920     |      +5.0        |       0.00260         |
> | **Qwen2.5-32B (Base)**| 3.4  | 16.8          | 34.3 | 4.5     | 63.2  | 7.8    | 21.7  |     –      |        –         |          –            |
> | **Guru-32B (Ours)**   | 34.9 | 29.3          | 50.6 | 7.6     | 82.0  | 12.6   | 36.2  |   7680     |     +14.5        |       0.00189         |
>
> **Guru-7B**: +5.0 points over base → 0.00260 points/GPU-hour
>
>
> **Guru-32B**: +14.5 points over base → 0.00189 points/GPU-hour
>
> This suggests that Guru-7B is more compute-efficient, delivering stronger reward improvement per GPU-hour. We will include this scaling and efficiency analysis in the revised paper. Future work will explore more model sizes to further improve accessibility and reduce compute requirements.
>
> ---
> > W6: While tasks like Zebra Puzzle and Logic Graphs suggest some temporal reasoning, the paper does not clearly evaluate long-horizon tasks. Please add or highlight benchmarks that explicitly test multi-step chain-of-thought or deductive planning.
>
> While we do not explicitly categorize tasks as long-horizon in the paper, many of our benchmarks inherently involve multi-step reasoning:
>
> (1) Several tasks, especially challenging math problems from math competitions, implicitly require multi-step deductive reasoning and symbolic manipulation.
>
> (2) Some of the synthetic logic tasks exhibit longer-horizon reasoning. For example, Zebra Puzzle and Graph Search typically require planning over 10+ steps, involving the chaining of constraints or path-finding over entity graphs.
>
> In the revised version of the paper, we will add a summary of the reasoning skills required for each task. We also see it as a valuable direction for future work to include more benchmarks that explicitly target long-horizon reasoning and deductive planning.
>
> In particular, future work could explore agentic tasks, where the LLM interacts with an external environment. Such settings provide a more structured framework for evaluating and improving long-horizon reasoning.

---

### Decision · Program_Chairs · 2025-09-18

**Decision:**

Accept (poster)

**Comment:**

This paper presents a collection of datasets designed to be use for RL for reasoning models.  Each dataset is filtered down and paired with a verifier for that domain. The paper shows cross-task transfer from RL on individual tasks, as well as strong performance from two trained models.

Strengths
1. The general ingredients of the dataset construction (scale, diversity, etc.) are strong
2. The released models achieve commendable performance and may be a useful starting point for other researchers due to their strength; these go beyond many released fine-tunes in research papers.

Weaknesses
1. The contribution here is primarily engineering: grouping together many datasets, training models of substantial size on them, etc. (However, it is good engineering!)

Discussion: the reviewers asked for ablations of filtering, SFT vs. RL, and other comparison experiments. Most significantly, xCcn and 5khS raised issues of how the results are presented compared to ORZ and lack of results on some datasets. The authors have now reported additional results, especially on MATH500, which I don't think substantially change the narrative.  As for potentially unfair conclusions to baselines: I would advise that the authors tend to focus on "non-aggregate" performance comparisons to models like ORZ. I agree it is unfair to fine-tune on additional datasets and declare victory on the basis of those datasets.

However, ultimately, I see this as a strong enough dataset effort and piece of engineering to be publishable.